# miR-23b-3p Modulating Cytoprotective Autophagy and Glutamine Addiction in Sorafenib Resistant HepG2, a Hepatocellular Carcinoma Cell Line

**DOI:** 10.3390/genes13081375

**Published:** 2022-08-01

**Authors:** Ramanpreet Kaur, Shruthi Kanthaje, Sunil Taneja, Radha K. Dhiman, Anuradha Chakraborti

**Affiliations:** 1Department of Experimental Medicine and Biotechnology, Postgraduate Institute of Medical Education & Research (PGIMER), Chandigarh 160012, India; ramanbiotech11@gmail.com (R.K.); shruthi.kanthaje@gmail.com (S.K.); 2Department of Hepatology, Postgraduate Institute of Medical Education & Research (PGIMER), Chandigarh 160012, India; drsuniltaneja@hotmail.com (S.T.); rkpsdhiman@hotmail.com (R.K.D.)

**Keywords:** hepatocellular carcinoma, sorafenib resistance, autophagy, glutamine addiction, miR-23b-3p

## Abstract

Background: Hepatocellular carcinoma (HCC) is the second most common malignancy with increasing cancer deaths worldwide. HCC is mainly diagnosed at its advanced stage, and treatment with FDA-approved sorafenib, the multikinase inhibitor drug, is advised. Acquired resistance against sorafenib develops through several pathways involving hypoxia, autophagy, high glycolysis, or glutaminolysis. Small non-coding RNAs, similar to microRNAs (miRNAs), are also known to affect sorafenib resistance in HCC. However, there is a lack of information regarding the significance of differentially expressed miRNA (if any) on autophagy and glutamine regulation in sorafenib-resistant HCC. Methods: The expression of autophagy and glutaminolysis genes was checked in both parental and sorafenib resistant HepG2 cell lines by real-time PCR. MTT and Annexin/PI assays were also performed in the presence of inhibitors such as chloroquine (autophagy inhibitor) and BPTES (glutaminolysis inhibitor). Next generation sequencing and in silico analysis were performed to select autophagy and glutamine addiction-specific microRNA. Selected miRNA were transfected into both HepG2 cells to examine its effect on autophagy and glutamine addiction in regulating sorafenib-resistant HCC. Results: Our in vitro study depicted a higher expression of genes encoding autophagy and glutaminolysis in sorafenib-resistant HepG2 cells. Moreover, inhibitors for autophagy (chloroquine) and glutaminolysis (BPTES) showed a diminished level of cell viability and augmentation in cell apoptosis of sorafenib-resistant HepG2 cells. NGS and real-time PCR demonstrated the downregulated expression of miR-23b-3p in sorafenib-resistant cells compared to parental cells. In silico analysis showed that miR-23b-3p specifically targeted autophagy through *ATG12* and glutaminolysis through *GLS1*. In transfection assays, mimics of miR-23b-3p demonstrated reduced gene expression for both *ATG12* and *GLS1*, decreased cell viability, and increased cell apoptosis of sorafenib-resistant HepG2 cells, whereas the antimiRs of miR-23b-3p demonstrated contrasting results. Conclusion: Our study highlights the cytoprotective role of autophagy and glutamine addiction modulated by miR-23b-3p (tumor suppressor), suggesting new approaches to curb sorafenib resistance in HCC.

## 1. Introduction

Hepatocellular carcinoma (HCC) is the second most common malignancy increasing cancer-associated mortality worldwide [1,2]. An age-standardized mortality rate of 6.8 for men and 5.1 for women per 100,000 people annually was reported in India [3]. Among several etiologic factors, including infection from hepatitis B and C viruses, alcohol, and aflatoxin, HBV infection is predominant in HCC pathogenesis in India [4]. Sorafenib is an FDA-approved first-line systemic therapy for advanced HCC. Sorafenib, a multikinase inhibitor for inhibiting Raf family kinases, vascular endothelial growth factor receptor (VEGFR), and platelet-derived growth factor receptor (PDGFR), resulting in enhanced angiogenesis, tumor proliferation, and apoptosis [5]. Most HCC patients show primary sorafenib resistance due to their genetic heterogeneity [6] or develop acquired sorafenib resistance from sustained drug exposure [7]. In acquired sorafenib resistance, several mechanisms are involved, including PI3K/Akt and JAK-STAT as compensatory pathways, hypoxia-inducible factors, transcription factors such as c-myc, and epithelial to mesenchymal transition factors including vimentin [8], autophagy, microRNAs (such as miR-21; miR-122), tumor microenvironments, or tumor metabolism [9,10].

The death-promoting effect of Sorafenib-induced autophagy has been reported by recent studies [11]. Although several studies have shown that autophagy has a protective role in HCC cells, the combination of autophagy inhibitors, including chloroquine, hydroxychloroquine, or autophagy-associated microRNAs, enhances the cytotoxic effects of sorafenib in HCC cells [12,13]. However, contrasting studies also show the cytotoxic role of autophagy [14]. Thus, the exact role of autophagy in sorafenib-resistant HCC is still controversial, and understanding its molecular mechanism is required to curb resistance.

Moreover, emerging evidence demonstrates that sorafenib is also involved in mediating the metabolic reprogramming (enhanced glycolysis and glutamine addiction) of HCC cells [15], which may develop drug resistance [16]. Exogenous glutamine is vital for cancer cells’ survival; hence, cancer cells show “glutamine addiction” [17]. In some human tumors, the upregulated expression of *GLS1* (glutaminase) is associated with advanced disease stages [18]. BPTES (bis-2-(5-phenylacetamido1,3,4-thiadiazol-2-yl)ethyl sulfide), the chemical inhibitor of *GLS1*, is well known for its tumor suppressive role in different preclinical models [19]. However, the role of glutaminase enhancing glutamine addiction in sorafenib-resistant HCC is yet to be defined. Hence, understanding the molecular mechanism of glutamine addiction in sorafenib-resistant HCC may also improve HCC treatment.

MicroRNAs (miRNAs), small (18–25 nucleotides) non-coding RNA molecules that regulate their targeted genes at the post-transcriptional level, are also involved in sorafenib resistance. Although miR-122 is the highest expressed miRNA in the liver, there are only a few reports on miRNAs targeting signaling pathways to influence sorafenib resistance and its related pathways [20]. These circulatory miRNAs are known as specific regulators that mediate any signaling pathway by targeting their specific genes. However, the influence of differentially expressed miRNAs in the modulation of autophagy and glutamine addiction to mediate sorafenib resistance in HCC is yet to be evaluated.

Here, we explored the role of autophagy and glutamine addiction in sorafenib-resistant HepG2 cells, the cellular model of HCC. Furthermore, we determined the significance of differentially expressed miRNAs associated with autophagy and glutamine addiction to curb sorafenib resistance and, thus, introduced new approaches to clinical HCC management.

## 2. Materials and Methodology

### 2.1. Chemicals

Minimum Essential Media (MEM) for cell culture and antibiotics were purchased from HiMedia (Chandigarh, India), Thermo Fisher Scientific (Waltham, MA, USA), and Sigma–Aldrich (St. Louis, MO, USA). Sorafenib was purchased from SignalChem Lifesciences Corporation (Canada). Chloroquine and Rapamycin were in lyophilized form and provided by the CYTO-ID^®^ Autophagy Detection Kit purchased from Enzo Life Sciences (Farmingdale, NY, USA). BPTES was purchased from Sigma–Aldrich Chemical Co. (St. Louis, MO, USA). Sorafenib (stock 1 mM), rapamycin (stock 10 µM), and BPTES (stock 20 µM) were prepared in DMSO, whereas chloroquine (stock 5 mM) was prepared in deionized water.

### 2.2. Cell Lines

Hepatocellular carcinoma (HCC) is the primary cancer of the liver. The HepG2 cell line is a liver cancer cell line derived from well-differentiated HCC patients and carries the wild-type p53 expression. Therefore, it more closely mimics HCC. HepG2 cell lines were cultured in MEM media in the presence of required conditions. Sorafenib-resistant HepG2 cells were already established in our lab [8]. HepG2 parental cells were treated with minimal dosage inhibitory concentration (IC-20) of sorafenib. Then, the dosage of sorafenib was increased continuously so that HepG2 cells could survive or adapt to low to high dosages of sorafenib until IC-50 achieved more than twofold resistant cells compared to parental cells [8].

Fold resistance = IC-50 value of resistant cells/IC-50 value of parental cells

Cultured cells were routinely monitored under an inverted light microscope (Olympus CKX41, Tokyo, Japan) and imaged using Olympus cellSens software for morphological characteristics.

### 2.3. RNA Isolation and Its Quantification, cDNA Synthesis, and qRT-PCR

Total RNA was extracted from parental and sorafenib-resistant HepG2 cells with Trizol reagent, as discussed in our previous study [8]. The relative mRNA level was calculated with the formula 2-^∆∆Ct^ [21], where ∆Ct = (Ct target gene − Ct internal control) and ∆∆Ct = (Ct sorafenib resistant cells − Ct parental cells). GAPDH was used for normalization (Table 1).

### 2.4. MTT Assay

Both parental and sorafenib-resistant HepG2 cells were cultured with 5000 cells per well in a 96-well plate (Corning Inc., Corning, NY, USA) to check cell viability. After 24 h incubation, the required concentration of treatments was added to fresh media. After 24 h, sterile 20 µL MTT (3-(4,5-Dimethylthiazol-2-yl)-2,5-Diphenyltetrazolium Bromide) was added to wells, and the plates were incubated for 4 h in a cell culture incubator at 37 °C with 5% CO_2_. Media along with MTT was removed, DMSO (200 µL) was added, and the plates were incubated for 15 min at RT in the dark to dissolve formazan crystals. The absorbance was measured at 570 nm with a microplate reader.

A dose response curve was plotted with % cell viability in the *Y*-axis and concentration of sorafenib or other chemical in the *X*-axis. Percentage of cell viability was measured by the formula:% Cell viability = ((Absorbance of test − Absorbance of Blank)/(Absorbance of control − Absorbance of Blank)) * 100

IC-50 values were then calculated from the dose response curve.

### 2.5. Annexin V/Propidium Iodide (PI)

Parental and sorafenib-resistant cells were treated with the required concentration of chemicals for 24 h and analyzed for apoptosis with a commercial Apoptosis Kit (Invitrogen, Waltham, MA, USA) [22]. Briefly, 1 × 10^6^ cells were trypsinized, washed with PBS, and centrifuged. Then, 100 μL of annexin binding buffer (1×) and 0.2 μL of Annexin V-FITC followed by 0.1 μL of propidium iodide (PI) were added to each tube except for the Annexin and PI tubes. The samples were incubated at room temperature for 15 min in the dark and supplemented with 150 μL of annexin binding buffer (1×) prior to analysis. The cells were checked by a flow cytometer (FACS Canto, Becton Dickinson, CA, USA) under 488 nm for fluorescein detection and a filter >600 nm for PI detection.

### 2.6. Green Autophagy Dye Assay

CYTO-ID**^®^** Autophagy Detection Kit (Enzo Life sciences) was used to detect the level of autophagy by flow cytometry in both parental and sorafenib-resistant HepG2 cells. The assay provides a rapid, quantitative approach to monitoring autophagic activity in vitro. The green fluorescent detection reagent becomes brightly fluorescent in autophagy vesicles. Flow cytometry calculated the level of green-emitting fluorescent probe corresponding to the number of green autophagy-stained cells.

### 2.7. Next Generation Sequencing (NGS)

The total RNA isolated from parental and resistant cells of HepG2 was used for microRNA (miRNA) analysis by next generation sequencing (NGS) in Medgenome Labs Pvt. Ltd., Bangalore, India. Illumina HiSeq2500 was used to perform miRNA NGS, and the data were mapped to the HG19 human genome database and miRBase 20 to obtain a list of deregulated miRNAs.

### 2.8. In Silico Analysis

In silico analysis was performed to determine microRNAs (miRNAs) as specific regulators targeting sorafenib-resistant pathways. Genes and their miRNA(s) targets were examined using three different databases, including miDRB (http://mirdb.org/, accessed on 16 May 2022) and miRWalk 2.0 predicted (http://zmf.umm.uni-heidelberg.de/apps/zmf/mirwalk2/generetsys-self.html, accessed on 16 May 2022) and validated (http://zmf.umm.uni-heidelberg.de/apps/zmf/mirwalk2/miRpub.html, accessed on 16 May 2022) databases. Additionally, we employed a set of four databases (miRWalk, miRanda, RNA22, and TargetScan) from the miRwalk predicted database and only considered targeted miRs predicted by these four databases [23]. The validated targeted miRNAs from miRwalk validated were considered [24]. Moreover, these miRNA(s) were reverse-analyzed as specific regulators of their targeted genes.

### 2.9. Transfection of miRNA Mimics and Antimirs

Mimics are commercially synthesized small double-stranded RNAs that upregulate miRNA activity, whereas antimiRs are commercially synthesized small single-stranded RNA molecules that specifically bind to endogenous miRNA molecules and downregulate microRNA activity. MirVana miRNA mimics, antimiRs, and scrambled miRNA (negative control) were purchased from Thermo Fisher Scientific Inc, Waltham, MA, USA. Transfection was performed using Lipofectamine RNAiMAX (Thermo Fisher Scientific Inc., Waltham, MA, USA) according to the manufacturer’s protocol. Briefly, Lipofectamine RNAiMAX reagent (3 µL) was added to 50 µL of MEM (without antibiotics). Simultaneously, 25 nM of the mimic/antimiR (siRNA) was added to 50 µL of MEM (without antibiotics). Lipofectamine RNAiMAX reagent and siRNA were added in a 1:1 ratio and incubated at RT for 5 min. For transfection, 2 × 10^5^ cells were seeded into a 24-well plate to obtain 60–80% confluence, and then 50 µL of the prepared mixture was added. Transfection was confirmed by qRT-PCR.

### 2.10. MiRNA Isolation; Polyadenylation, cDNA Synthesis Poly A Tailed miRNA, qRT-PCR for miRNAs

The extracellular (as miRNAs secreted in cell culture media) miRNAs were isolated by the Trizol reagent, as discussed in our previous study [8]. Isolated miRNA quantified by an infinite M200PRO microplate reader (Tecan, Männedorf city, Switzerland). The absorbance 260/280 of >1.8 was considered an acceptable value. Each miRNA molecule was polyadenylated using the miRNA cDNA synthesis kit (Agilent Technologies, Santa Clara, CA, USA) according to the manufacturer’s instructions. Finally, the prepared cDNA was stored at −20 °C until further use. Then, the extracellular (as secreted miRNAs in cell culture media) miRNA expression levels were quantified by qRT-PCR performed by Lightcycler^®^ 480 (Roche, Switzerland) using SYBR Green I master mix (Thermo Fisher Scientific Inc., Waltham, MA, USA). qRT-PCR was performed by Lightcycler^®^ 480 (Roche, Switzerland) using SYBR Green I master mix (Thermo Fisher Scientific Inc., Waltham, MA, USA). The forward primer of miRNA was the miRNA sequence and synthesized by Sigma–Aldrich. The reverse primer is the universal reverse primer (URP) provided by the kit (Agilent Technologies, Santa Clara, CA, USA). The miRNA expression was normalized using U6, a small nuclear RNA, acting as the internal control. A total of 1 µg of total RNA was used for intracellular miRNA assessments and 150 ng of miRNA for extracellular. The reaction mixture contained cDNA (1 µL), 0.3125 µM primer (forward and reverse), 1X SYBR Green Mix, and nuclease-free water for volume makeup (up to 10 µL). In all cases, initial denaturation occurred at 95 °C, for 5 min, and final melting (5 °C + annealing temperature; 1 min).

The expression levels of miRNA in test vs. control were determined by fold change using the formula (2^−ΔΔCt^) [23].

Primer sequence of miRNAs; FP: Forward Primer; RP: Reverse Primer (Table 2).

### 2.11. Statistical Analysis

The required experiments were conducted three times independently. The statistical calculations were performed using GraphPad Prism 6.01 (GraphPad Software Inc., San Diego, CA, USA) software for Windows. ANOVA (for multiple comparisons), Student’s *t*-test, and Mann–Whitney U test (two-group comparison) were used as required. The experimental data were shown as mean ± SEM, and *p* < 0.05 was considered statistically significant.

## 3. Results

### 3.1. Autophagy and Glutamine Addiction as Cytoprotective in Sorafenib-Resistant HepG2 Cells

Both HepG2 parental HepG2 (P) and HepG2 sorafenib resistant HepG2 (R) cells [8] were stored at −80 °C and were cultured for required experiments (Appendix A). The autophagy-related genes (ATGs) expression was checked; in HepG2 (R) cells, the expression of *ATG7* was upregulated with fold change 1.513 ± 0.2684, whereas the levels of *Beclin1*, *LC3II*, and *ATG12* significantly increased with fold change 1.678 ± 0.05773; 1.733 ± 0.08665; 1.686 ± 0.06741, respectively, compared with HepG2 (P) cells (Figure 1, Appendix A).

Furthermore, with increasing concentrations (50, 80, and 100 nM) of rapamycin (autophagy initiator), the cell viability of HepG2 (R) was high. By contrast, increasing concentrations (50, 80, and 100 µM) of chloroquine (autophagy inhibitor) significantly decreased (*p* = 0.0034 at 80 µM (IC_50_) the cell viability of resistant cells. Both sorafenib (4 µM; IC_50_) and chloroquine (80 µM; IC_50_) also significantly (*p* = 0.0007) reduced the cell viability of HepG2 (R) cells (Figure 2A,B). Afterward, the presence of chloroquine showed an increased percentage of apoptosis 50.6 ± 0.9815 in HepG2 (R) (Figure 2C,D), suggesting the cytoprotective role of autophagy.

Initially, we checked the expression of the oncogenic gene *c-myc*, which was significantly (*p* = 0.0011) higher in HepG2 (R) cells than in HepG2 (P) cells with a fold change of 2.533 ± 0.1822 (Appendix A). The increase in sorafenib concentrations (2 µM and 4 µM) significantly (*p* = 0.0015) enhanced the expression of *c-myc* with 2.267 ± 0.2942 and 4.8 ± 0.2309-fold changes respectively (Appendix A). The expression of *c-myc* with glutaminolysis through targeting its vital enzyme glutaminase (*GLS1*) correlated with HepG2 (R) cells. The *GLS1* expression was significantly (*p* = 0.0008) higher in HepG2 (R) than HepG2 (P), with a fold change of 2.797 ± 0.196 (Figure 3).

Both HepG2 (P) and HepG2 (R) cells were cultured in MEM media without exogenous glutamine. Inverted microscopy has shown elongated morphology with a decreased proliferation rate in both HepG2 cells. However, the proliferation rate was lower in HepG2 (R) cells than in HepG2 (P) cells (Figure 4A). The MTT assay also showed resistant cells without exogenous glutamine [R (-glut)] that exhibited significantly (*p* < 0.0001) reduced cell viability (17.52 ± 1.613) compared to resistant cells with glutamine (47.27 ± 1.295) (Figure 4B).

The green autophagy dye-stained vesicles (markers of autolysosomes and autophagic compartments) were also examined using a CYTO-ID Autophagy detection kit. The increased autophagic flux was directly proportional to the increased number of green autophagy dye-stained vesicles detected through green emitting fluorescence using flow cytometry. The number of green autophagy dye-stained vesicles was significantly (*p* = 0.0153) higher in resistant cells than in parental cells. However, in the absence of exogenous glutamine sorafenib-resistant cells [R (-glut)], the number of green autophagy dye0stained vesicles significantly (*p* = 0.0321) increased with a difference of 224 ± 69.42 compared to the presence of glutamine (Figure 5A,B).

We performed an MTT assay using BPTES (glutaminase inhibitor), where the cell viability of BPTES-treated sorafenib-resistant cells was significantly (*p* < 0.0001) decreased compared to non-treated resistant cells (Figure 6A). The BPTES (300 nM; IC_50_) significantly (*p* = 0.0010) increased cell apoptosis (10.57 ± 1.216-fold higher) of resistant cells than in non-treated HepG2 cells (Figure 6B,C).

### 3.2. Autophagy and Glutamine Addiction-Specific microRNAs and their Target Genes in Sorafenib-Resistant HCC Cell Lines

The NGS data revealed 157 upregulated and 86 downregulated miRNAs in sorafenib-resistant compared to parental cells (Figure 7A). The in silico analysis using miRDB, the miRNAs target prediction online database, revealed the miR-23b-3p as autophagy-specific miRNA with a target score of 82 for *ATG12*, among other autophagy genes (*ATG7*, *Beclin1*, *LC3II*, and *ATG 12*) (Appendix A). The higher target score specifies the real prediction of the selected miRNA (http://mirdb.org/faq.html, accessed on 16 May 2022). The miR-23b-3p was also observed to target *GLS1* (the vital gene of glutamine addiction) with a target score of 90 (Appendix A). Furthermore, miRWalk, the online database including both predicted and validated targeted miRNAs to genes and combined with other databases (miRWalk, miRanda, RNA22, and Targetscan), also confirmed miR-23b-3p as miRNA targeting *ATG12* (Appendix A). Similarly, for glutamine addiction using a combination of different databases (miRWalk, miRanda, RNA22, and Targetscan), miR-23b-3p was selected as miRNA targeting *GLS1* [(Appendix A). Furthermore, in silico analysis using miRDB, miRWalk predicted and validated also revealed miR-23b-3p as a common miRNA targeting autophagy through *ATG12* and glutamine addiction through *GLS1* (Figure 7B,C).

The expression analysis of *ATG12* and *GLS1* was also evaluated, where the presence of increasing sorafenib concentrations ranging from 2 to 6 µM and the expression of *ATG12* significantly (*p* = 0.0226) increased with a fold change of 4.59 ± 0.6566 (Figure 8A). Similarly, the expression of *GLS1* also significantly (*p* = 0.0014) increased with a fold change of 6.13 ± 0.4131 (Figure 8B).

The extracellular expression of miR-23b-3p (miR-23) was evaluated in the presence of increasing concentrations of sorafenib (4 and 6 µM). The increasing concentration of sorafenib decreased the miR-23 extracellular expression. Higher concentrations of sorafenib (6 µM) significantly (*p* = 0.0013) decreased the extracellular expression of miR-23 with a fold change of 0.3027 ± 0.05773 compared to resistant cells without sorafenib treatment (Figure 9).

### 3.3. Functional Analysis of miR-23b-3p

Mimics, as well as antimiRs of miR-23b-3p, were purchased from Thermo Fisher Scientific, Waltham, MA, USA. The transfection of both mimics and antimiRs of miR-23b-3p was executed in HepG2 cells to analyze the functional perspective of microRNA (Appendix A). To standardize the maximum efficient transfection concentrations of mimics and antimiRs, a cell viability assay with increasing concentrations (ranging from 10 to 30 nM) of mimics and antimiRs was performed at 30 nM (denoted further as 30 (M) and 30 (A) resp.). AntimiR [30 (A)] enhanced the cell viability at a maximum of both HepG2 cells, whereas the mimic [30 (M)] significantly (*p* = 0.0001) decreased the cell viability with a mean difference of 77.43 ± 4.741 in HepG2 (R) cells compared to non-treated resistant HepG2 cells) (Appendix A). Hence, we selected mimic and antimiR concentrations of 30 nM for their transfection assays.

The autophagy and glutamine-specific miR-23b-3p expression was evaluated in the presence of its enhancer (mimic) or inhibitor (antimiR) in HepG2 (P) cells where scrambled miRNA was used as the negative control (NC). The transfection of mimic (miR-23m) significantly (*p* = 0.0127) increased miR-23b-3p (miR-23) expression compared to the negative control (NC). However, the transfection of antimiR (miR-23a) significantly (*p* = 0.0234) decreased miR-23b-3p expression compared to the negative control (Appendix A). Furthermore, the stability of the transfected mimic (30 nM) and antimiR (30 nM) was checked at periods ranging from 24 to 48 to 72 h. The transfection of both mimic and antimiR was found to be significantly (*p* = 0.0282) stable up to 72 h (Appendix A).

The antimiR (miR-23a) increased *ATG12* expression, whereas the presence of miR-23b-3p mimic (miR-23m) (30 nM) significantly (*p* = 0.0031) decreased the expression of *ATG12* in resistant HepG2 cells compared to non-transfected resistant cells (Figure 10A). Similarly, the antimiR (miR-23a) also increased *GLS1* expression, whereas *GLS1* expression was also significantly (*p* = 0.0342) decreased in sorafenib-resistant HepG2 cells transfected with miR-23b-3p mimic (miR-23m) (30 nM) compared to resistant cells without miR-23m transfection (Figure 10B).

The effects of mimics and antimiRs on the IC-50 value of sorafenib were evaluated by the transfection of both miR-23m and miR-23a in HepG2 (R) cells. The transfection of miR-23a enhanced the proliferation rate, whereas the presence of miR-23m decreased the proliferation rate in HepG2 (R) cells compared to non-treated sorafenib-resistant cells (Figure 11A,B). The significance of miR-23b-3p mimic (miR-23m) and antimiR of miR-23b-3p (miR-23a) was validated in both parental and sorafenib-resistant HepG2 cells by the Annexin/PI assay. The presence of miR-23b-3p mimic (miR-23m) in sorafenib-resistant HepG2 cells significantly (*p* = 0.0224) increased cell apoptosis percentage with a mean difference of 6.367 ± 1.761 compared to sorafenib-resistant cells without mimic transfection (Figure 11C,D).

## 4. Discussion

HCC is one of the predominant causes of cancer-associated deaths [2]. The FDA-approved oral drug sorafenib, a multikinase inhibitor, is administered for advanced HCC [25,26]. In India, the sorafenib treatment is easily tolerated and increases overall survival rate by three months [27]. However, acquired resistance limits the efficiency of sorafenib in HCC patients [7]. The main factors for acquired resistance against sorafenib are autophagy, altered metabolic reprogramming, and differentially expressed microRNAs [9,10]. HepG2 cells, the cellular model of HCC, are used to find cytotoxicity levels and cellular and drug metabolism. Initially, we investigated the significance of autophagy in sorafenib-resistant HepG2 cells. The higher expression of autophagy-related genes (ATGs): *ATG7*, *ATG12*, *Beclin1*, and *LC3II* in HepG2 (R) cells compared to HepG2 (P) cells indicated the protective role of autophagy in sorafenib-resistant HepG2 cells. Moreover, decreased cell proliferation and increased cell apoptosis of sorafenib-resistant HepG2 cells in the presence of chloroquine (the autophagy inhibitor) also confirmed the protective role of autophagy in sorafenib-resistant HCC. Similarly, studies performed by Manov et al. [28] and Shimizu et al. [12] also discussed the protective role of autophagy in HCC. By contrast, the cytotoxic role of autophagy was discussed in HCC (only) by Tai et al. [11] and Zhai et al. [14]. However, our study is the first to demonstrate the cytoprotective role of autophagy in HepG2 (R) cells.

Altered tumor metabolism, including high glycolysis and glutaminolysis, is also crucial in cancer resistance. We demonstrated a higher glutaminase expression (*GLS1*) in HepG2 (R) cells. In ovarian cancer progression, the higher expression of *GLS1*, the vital gene of glutaminolysis, has also been discussed [29]. In the present study, the absence of exogenous glutamine or the presence of BPTES (glutaminase inhibitor) reduced cell viability and increased cell apoptosis of sorafenib-resistant cells, validating the cytoprotective significance of glutamine addiction in HepG2 (R) cells. The CYTO-ID**^®^** Autophagy Detection Kit evaluated higher numbers of autophagy-stained vesicles, suggesting that detrimental glutamine might enhance autophagy pathways in sorafenib-resistant HCC, which is similar to a study performed by Zhu et al. [30].

We also correlated the importance of differentially expressed microRNAs (miRNAs) in association with autophagy and glutamine addiction in sorafenib-resistant HCC. A recent study showed that the *SNGH16* regulates autophagy through miR-23b-3p and enhances sorafenib resistance [31]. Our next generation sequencing (NGS) and in silico analyses evaluated miR-23b-3p as specific miRNA targeting *ATG12* (autophagy) and *GLS1* (glutamine addiction). The association between miR-23b-3p and *ATG12* has been discussed in gastric cancer cells [32]; metabolic reprogramming has also been described in osteosarcoma [33]. However, the influence of miR-23b-3p’s association with *ATG12* and autophagy or *GLS1* and glutamine addiction on sorafenib resistance in HCC has not been discussed. Our NGS data and extracellular expression demonstrated downregulated levels of miR-23b-3p in HepG2 (R) cells, suggesting the tumor-suppressing role of miR-23b-3p. Similarly, the downregulated expression of miR-23b-3p as a predictor has been discussed in HCC progression [34]. Furthermore, the association between sorafenib treatments in HCC patients and the longitudinal variation of miR-23b-3p expression has been discussed in recent research [35]. However, for the first time, our study highlights the significance of tumor-suppressing miR-23b-3p in modulating cytoprotective autophagy through *ATG12* and glutamine addiction through *GLS1* in sorafenib-resistant HCC.

## 5. Conclusions

Our study determined the cytoprotective role of both autophagy and glutamine addiction in HepG2 (R) cells. The NGS and in silico analysis confirmed miR-23b-3p as differentially expressed miRNA targeting cytoprotective autophagy through *ATG12* and glutamine addiction through *GLS1* in sorafenib-resistant HepG2 cells. Transfection assays validated the tumor-suppressing effect of miR-23b-3p and modulated the expression of both *ATG12* and *GLS1* in sorafenib-resistant HepG2 cells, curtailing sorafenib resistance in HCC.

## Figures and Tables

**Figure 1 genes-13-01375-f001:**
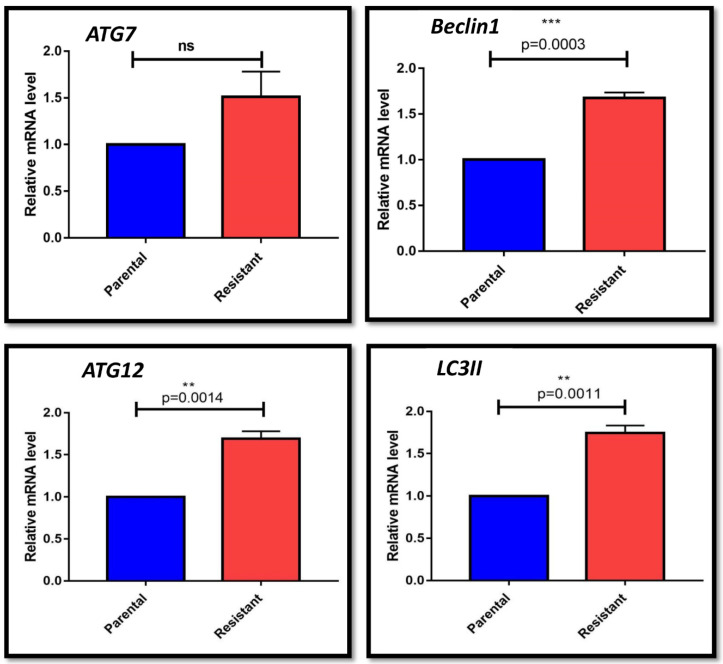
Bar graph representation of expression analysis of autophagy genes (Parental: Parental HepG2 cells; Resistant: Sorafenib resistant HepG2 cells) (ns: non-significant, *p* > 0.05; ** *p* ≤ 0.01; *** *p* ≤ 0.001).

**Figure 2 genes-13-01375-f002:**
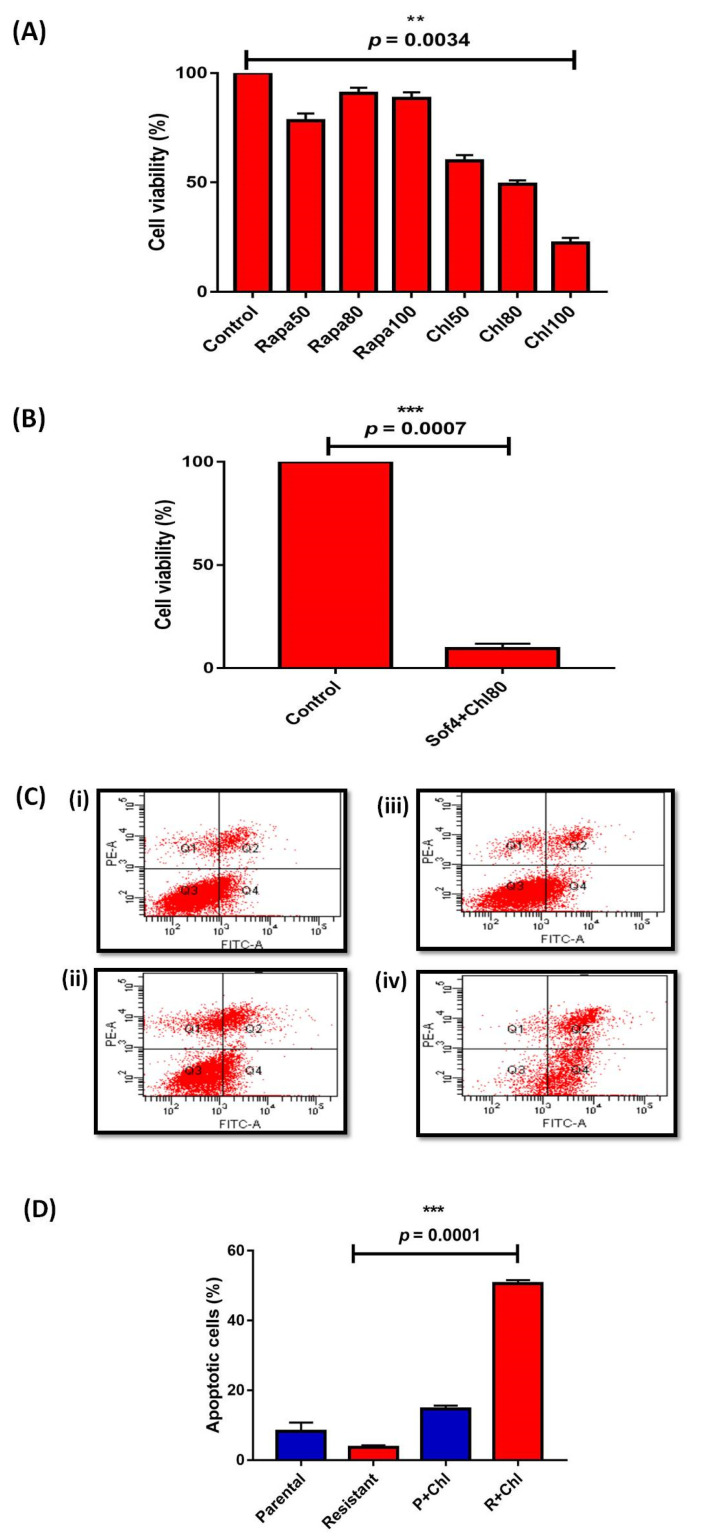
MTT assay showing (**A**) Percentage of cell viability in sorafenib resistant treated with different concentrations [(Rapa: Rapamycin (nM); Chl: Chloroquine (µM)] (**B**) Percentage of cell viability in presence of combination of sorafenib (Sof; IC_50_: 4 µM) and chloroquine (Chl; IC_50_: 80 µM) (**C**) Annexin V/PI assay showing cell apoptosis (i) Parental HepG2 cells (ii) sorafenib resistant HepG2 cells (iii) Parental+chloroquine (80 µM) [P+Chl] (iv) sorafenib resistant+chloroquine 80 µM [R+Chl] (**D**) Bar graph representation of percentage of apoptotic cells in HepG2 parental and resistant cells (** *p* ≤ 0.01; (*** *p* ≤ 0.001)).

**Figure 3 genes-13-01375-f003:**
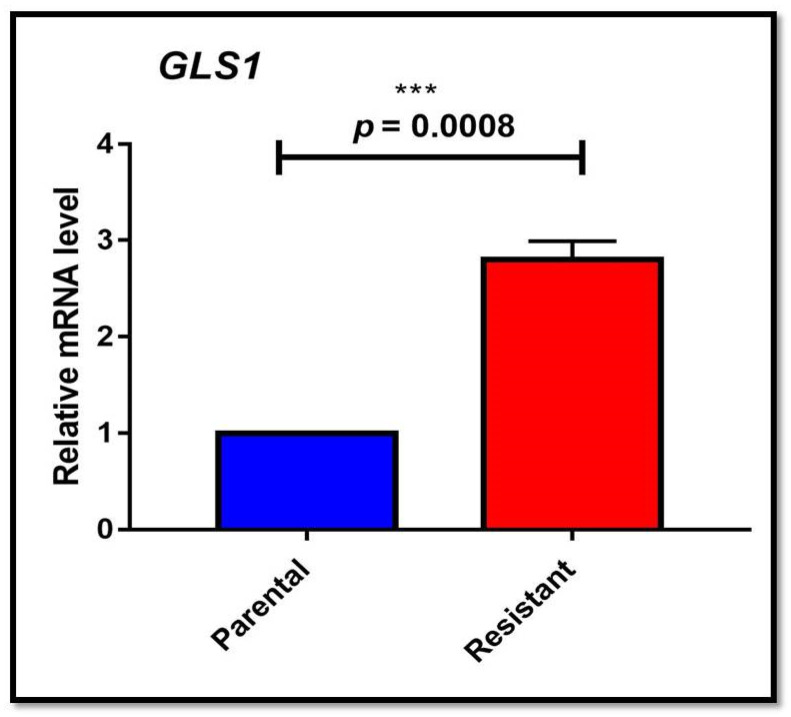
*GLS1* (glutaminase) expression analysis in parental and sorafenib resistant HepG2 cells (*** *p* ≤ 0.001).

**Figure 4 genes-13-01375-f004:**
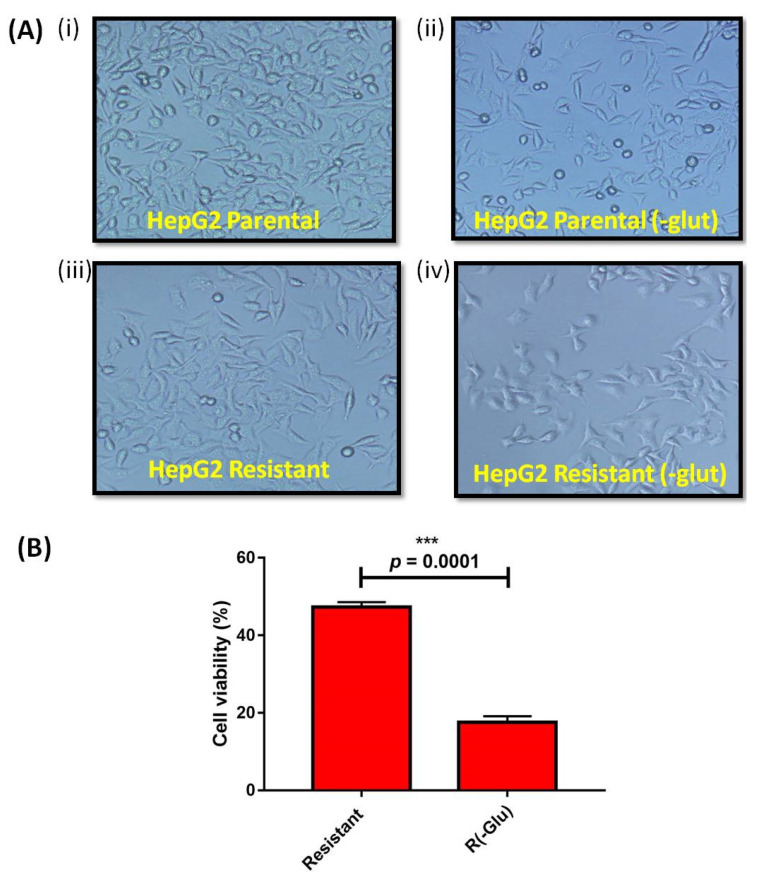
MTT assay showing cell viability (**A**) Morphological analysis (**B**) Bar graph representation of cell viability percentage in Resistant: sorafenib resistant HepG2 cells; R(-glut): sorafenib resistant HepG2 cells without exogenous glutamine (*** *p* ≤ 0.001).

**Figure 5 genes-13-01375-f005:**
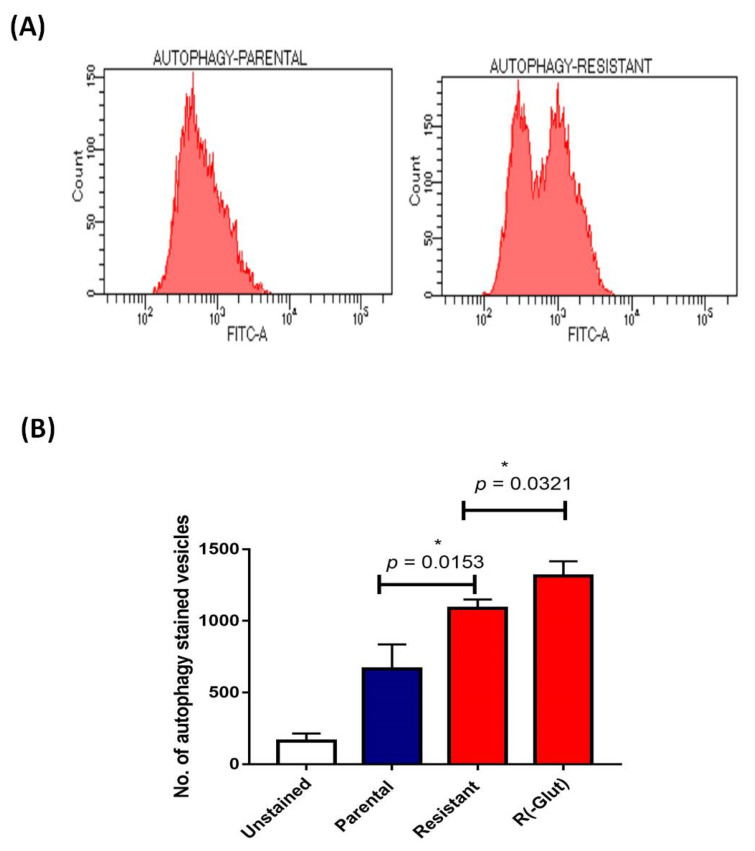
(**A**) Flow cytometry analysis of autophagy dye stained vesicles in parental and sorafenib resistant HepG2 cells (**B**) Bar graph representation of number of autophagy stained vesicles in HepG2 cells [R(-glut); sorafenib resistant HepG2 cells without exogenous glutamine] (* *p* ≤ 0.05).

**Figure 6 genes-13-01375-f006:**
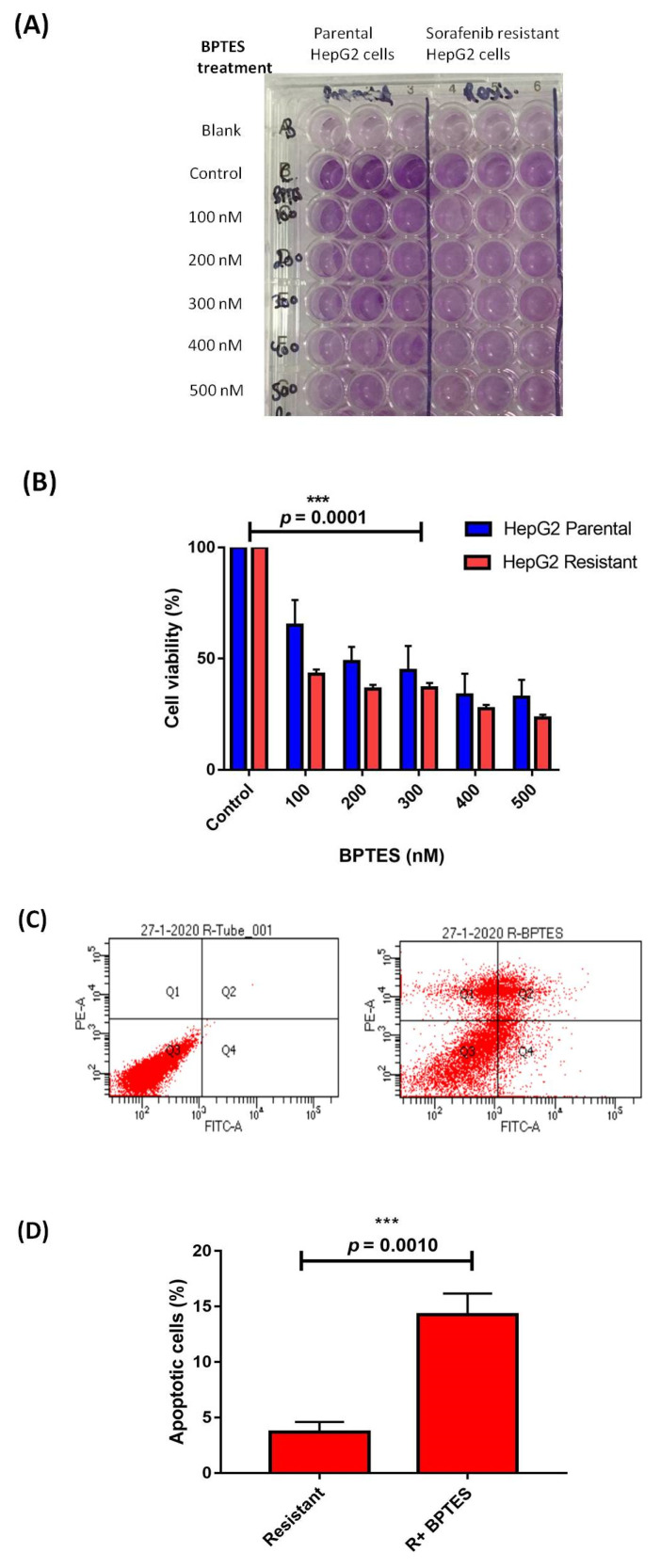
The cell viability (%) in parental and sorafenib resistant HepG2 cells by MTT assay in presence of increasing concentration of BPTES (nM) (**A**) 96-well plate (**B**) Bar graph representation of cell viability (**C**) Flow cytometry showing cell apoptosis (%) by Annexin V/PI analysis. (**D**) Bar graph representation of percentage of apoptotic cells in presence of BPTES (300 nM) in sorafenib resistant HepG2 cells (*** *p* ≤ 0.001). (**A**) The cell viability (%) in parental and sorafenib-resistant HepG2 cells by MTT assay in the presence of increasing concentrations of BPTES (nM). (**B**) Bar graph representation of cell viability. (**C**) Flow cytometry showing cell apoptosis (%) by Annexin V/PI analysis. (**D**) Bar graph representation of apoptotic cell percentage in the presence of BPTES (300 nM) in sorafenib-resistant HepG2 cells.

**Figure 7 genes-13-01375-f007:**
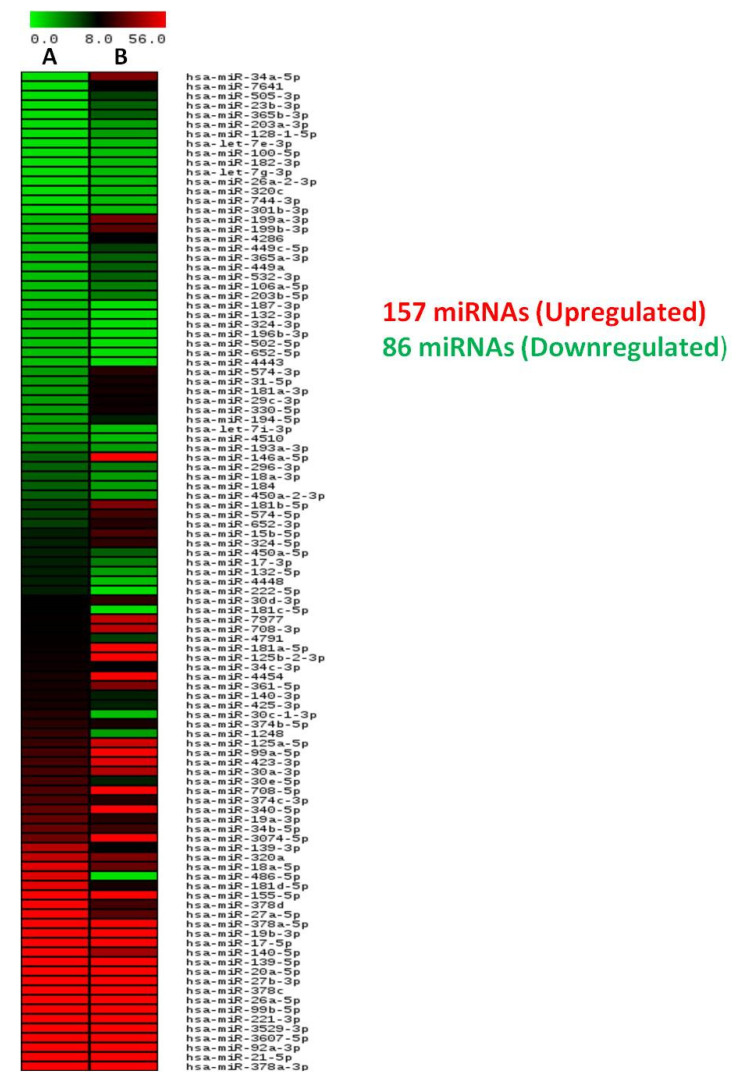
Next generation sequencing showing a heat map of differentially expressed microRNAs (miRNAs) in (**A**) Parental HepG2 cells; (**B**) sorafenib-resistant HepG2 cells; (**C**) differentially expressed miR-23b-3p in HepG2 (R) compared to HepG2 (P) by NGS; (**D**) Venn diagram representing commonly identified miR-23b-3p from three different databases targeting both autophagy and glutamine addiction, i.e., autophagy and glutamine addiction-specific microRNA (miR-23b-3p).

**Figure 8 genes-13-01375-f008:**
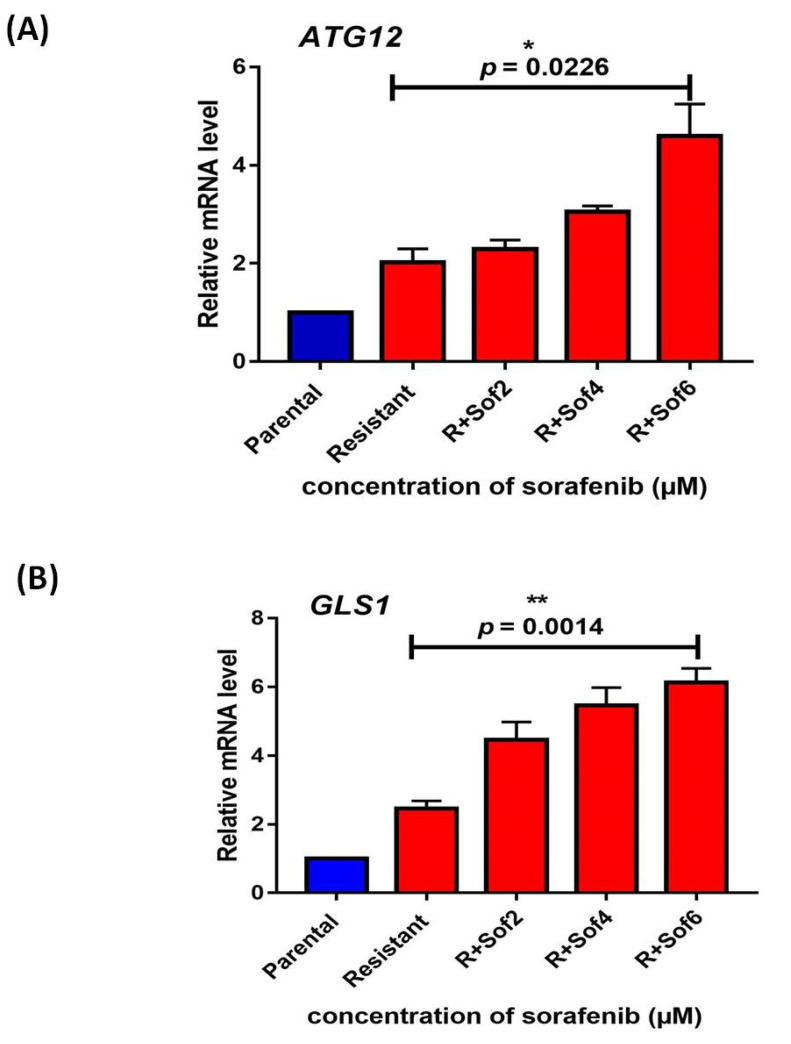
(**A**) *ATG12* expression analysis (**B**) *GLS1* expression analysis in parental and sorafenib resistant HepG2 cells [R; sorafenib resistant HepG2 cells; Sof: sorafenib with 2 µM; 4 µM; 6 µM] (* *p* ≤ 0.05); (** *p* ≤ 0.01).

**Figure 9 genes-13-01375-f009:**
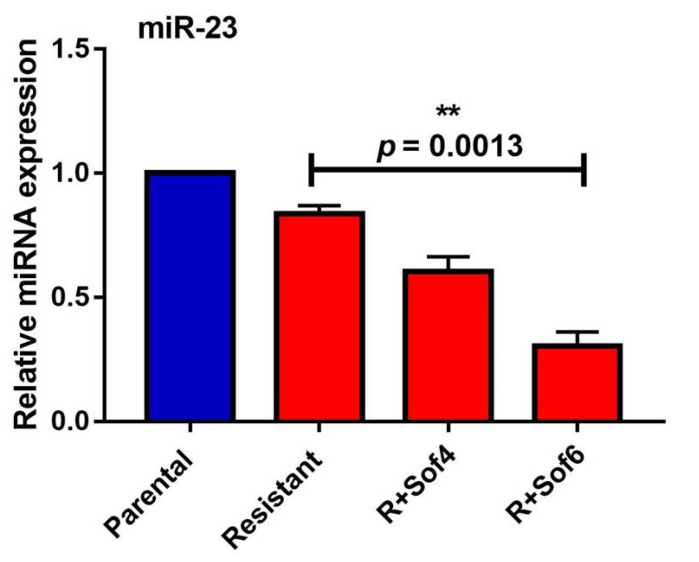
The extracellular expression analysis of miR-23b-3p (miR-23) in parental and sorafenib resistant HepG2 cells (** *p* ≤ 0.01).

**Figure 10 genes-13-01375-f010:**
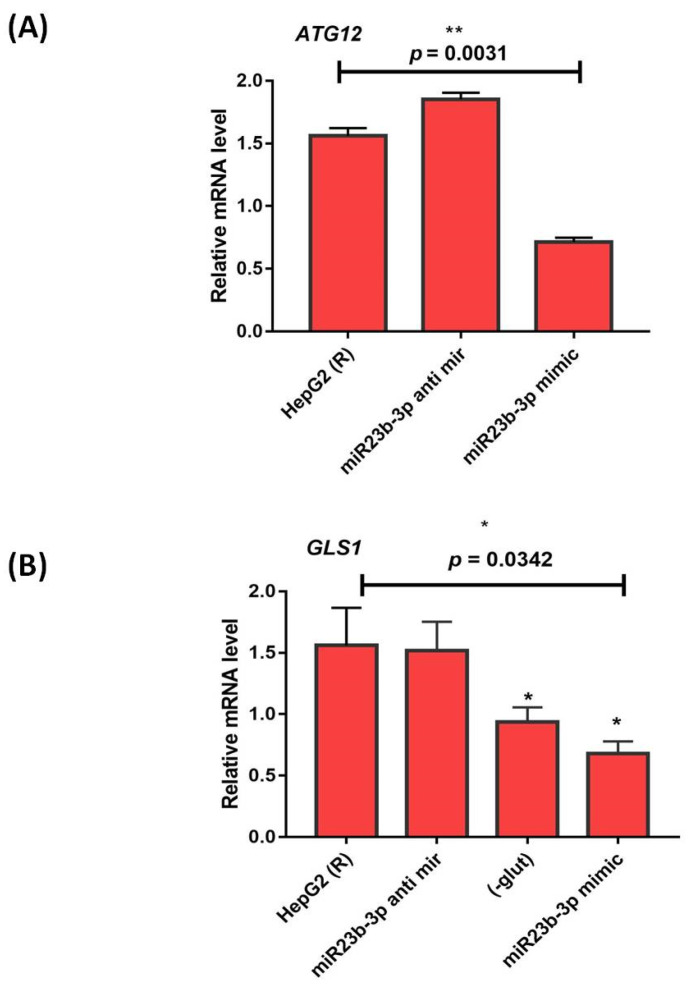
(**A**) ATG12 expression analysis (**B**) GLS1 expression analysis in presence of miR-23b-3p mimic (miR-23m) and antimiR (30 nM) in sorafenib resistant HepG2 cells [R; sorafenib resistant HepG2 cells] (* *p* ≤ 0.05); (** *p* ≤ 0.01).

**Figure 11 genes-13-01375-f011:**
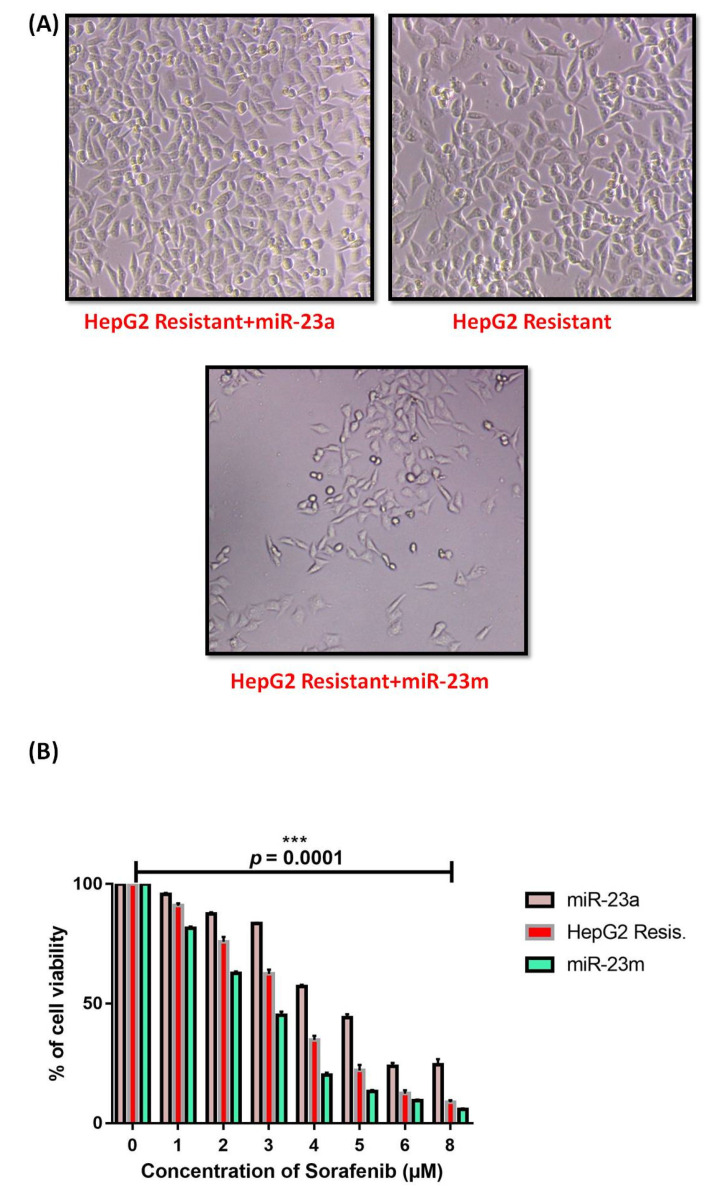
Effects of miR-23b-3p mimic (miR-23m) and antimiR (miR-23a) on sorafenib IC_50_ in sorafenib-resistant HepG2 cells. (**A**) Microscopic analysis. (**B**) The cell viability (%) and IC-50 of sorafenib in presence of miR-23b-3p mimic (miR-23m) and miR-23b-3p antimiR (mir-23a) in sorafenib resistant HepG2 cells by MTT assay (**C**) Flow cytometric analysis showing the percentage of cell apoptosis by Annexin V/PI analysis: (i) parental HepG2 cells; (ii) parental HepG2 cells + miR-23a; (iii) parental HepG2 cells + miR-23m; (iv) resistant HepG2 cells; (v) resistant HepG2 cells + miR-23a; (vi) resistant HepG2 cells + miR-23m [miR-23b-3p mimic (miR-23m); miR-23b-3p antimiR (miR-23a)]. (**D**) Bar graph representation of apoptotic cells (%) in presence of miR-23b-3p mimic (miR-23m) in HepG2 cells (*** *p* ≤ 0.001); (* *p* ≤ 0.05).

**Table 1 genes-13-01375-t001:** Primer sequences of genes.

Gene	Primer Sequence	Product Size (bp)	Thermal Conditions
*ATG7*	FP: ATGGTGCTGGTTTCCTTGCT RP: CTGCTACTCCATCTGTGGGC	158 bp	95 °C-10 s; 54 °C-15 s; 72 °C-20 s
*ATG12*	FP: GCGAACACGAACCATCCAAG RP: CACGCCTGAGACTTGCAGTA	189 bp	95 °C-10 s; 54 °C-15 s; 72 °C-20 s
*Beclin1*	FP: ACCGTGTCACCATCCAGGAA RP: GAAGCTGTTGGCACTTTCTGT	231 bp	95 °C-10 s; 54 °C-15 s; 72 °C-20 s
*LC3II*	FP: GATGTCCGACTTATTCGAGAGC RP: TTGAGCTGTAAGCGCCTTCTA	167 bp	95 °C-10 s; 59 °C-15 s; 72 °C-20 s
*c-MYC*	FP: TAGTGGAAAACCAGCCTCCC RP: CCGAGTCGTAGTCGAGGTCA	81 bp	95 °C-10 s; 56 °C-15 s; 72 °C-20 s
*GLS1*	FP: TCCCCAAGGACAGGTGGAA RP: ACGGTTTGATTTTCCTTCCCG	142 bp	95 °C-10 s; 64 °C-15 s; 72 °C-20 s
*GAPDH*	FP: CCATCTTCCAGGAGCGAGA RP: GGTCATGAGTCCTTCCACGAT	305 bp	95 °C-10 s; 60 °C-15 s; 72 °C-20 s

**Table 2 genes-13-01375-t002:** Primer Sequences of microRNA.

miRNA	Primer Sequence
hsa-miR-23b-3p	FP-ATCACATTGCCAGGGATTACCAC
U6	FP-CGCTTCGGCAGCACATATACTAA RP-TATGGAACGCTTCACGAATTTGC

## Data Availability

Not applicable.

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
