# Peer review of "miR-23b-3p Modulating Cytoprotective Autophagy and Glutamine Addiction in Sorafenib Resistant HepG2, a Hepatocellular Carcinoma Cell Line"

_genes, 2022, doi:10.3390/genes13081375_

Round 1

Reviewer 1 Report

Hepatocellular carcinoma (HCC) is the second leading cause of cancer related mortality, worldwide. Sorafenib is a multi-kinase inhibitor, approved by the FDA for treating advanced cases of HCC. However, acquired resistance against sorafenib develops through upregulation of  several pathways including autophagy, or glutaminolysis. Thus, metabolic reprogramming can drive the resistance to sorafenib, but the exact molecular mechanisms driving  and mediating this effect have not  been revealed. In the current work Kaur et al show that  the small RNA -mirR-23b-3p modulates the sensitivity of HCC cells to sorafenib through targeting ATG12 and the downregulation of autophagy, and parallel targeting of GLS1 and downregulation of  glutaminolysis. Conversely, downregulation of mirR-23b-3p itself leads to the upregulation of the two processes and to the acquirement of a resistance to sorafenib by HCC cells. These are very interesting findings that progress our knowledge and understanding in the cancer resistance field. The results  are presented in a clear and convincing figures with sound statistical analysis, except for Figure 11B which lacks an adequate statistical evaluation. Unfortunately, the Discussion section is rather superficial and lacks some depth. The authors should try and explain for example how would upregulation of glutamonolysis endow HCC cells with resistance to sorafenib. This should be based on the regulatory pathways which are known to be affected sorafenib.

Author Response

Thank you for your valuable comments. We have modified the manuscript as suggested by reviewers. All the queries have been highlighted in the text.

First Reviewer:

Comment 1.    Figure 11B lacks an adequate statistical evaluation

We agree with Reviewer’s comment. As per your suggestion, the statistical evaluation has been done. Please see revised result (Figure 11B) is the representation of cell viability and IC50 value of sorafenib in presence of transfection of both miR-23m and miR-23a in sorafenib resistant HepG2 cells. The transfection of antimiR of miR-23b-3p (miR-23a) enhanced the proliferation rate of sorafenib resistant cells whereas the presence of mimic of miR-23b-3p (miR-23m) in sorafenib resistant HepG2 cells decreased the proliferation rate of HepG2 cells in comparison to non treated sorafenib resistant cells. Hence, IC-50 value of sorafenib was increased with miR-23a transfection whereas IC-50 value of sorafenib was reduced with transfection of miR-23m in comparison to non-treated cells.

Comment 2.   Discussion section is rather superficial and lacks some depth.  

Discussion has been modified as highlighted (Page 18; Line no. 9-13; 31)

Comment 3.    How would upregulation of glutaminolysis endow HCC cells with resistance to sorafenib

Upregulation of Glutaminolysis results in increase in expression of its related genes. Here, we have also evaluated the expression of glutaminase (GLS1); the vital gene of glutaminolysis which showed an enhanced expression in sorafenib resistant HepG2 cells as compared to parental HepG2 cells. Thus, indicating the effect of glutaminolysis on sorafenib resistant cells through GLS1.

Reviewer 2 Report

Drug resistance through cell autophagy is a very interesting subject with clinical relevance.  Downregulated expression of miR-24 23b-3p in sorafenib resistant HepG2 cells is has been described recently (Zhao Jing et al. 2020. SNGH16 regulates cell autophagy to promote Sorafenib Resistance through suppressing miR-23b-3p via sponging EGR1 in hepatocellular carcinoma). Please add this reference in the background /discussion. 

Author Response

Thank you for your valuable comments. We have modified the manuscript as suggested by reviewers. All the queries have been highlighted in the text.

Second Reviewer:

Comment 1.    To add reference in the discussion

The reference has been discussed and added to the section (Highlighted Page 19; Line no. 4).

Reviewer 3 Report

Comments on the manuscript -miR-23b-3p Modulating Cytoprotective Autophagy and Glutamine Addiction in Sorafenib Resistant Hepatocellular Carcinoma Cells by the Authors: Ramanpreet Kaur, Shruthi Kanthaje, Sunil Taneja, Radha K Dhiman, Anuradha Chakraborti

I appreciated very much this article by Ramanpreet Kaur; the authors have raised very interesting issues in the field of sorafenib resistance in HCC and have pursued their specific tasks outlined in a sharp experimental design. The experimental data obtained clearly sustain the contention that miR-23b-3p plays its biological role as tumour suppressor in HCC. In particular the authors have shown the ability of this miR to modulate the cytoprotective autophagy and glutamine addiction in sorafenib resistant HCC cells. I am surprised that the authors have chosen the HepG2 cells, because, as they mention in the text, these are well differentiated HCC cells. Usually sorafenib is used for advanced HCC, that very seldom is a differentiated HCC. The authors should comment this point in the text of their article. The various experimental tasks have been accomplished by using several techniques. In the Section of Materials and Methods I did not find the description of miR mimics and antagomiR transfection in HCC cells. Further the authors should report the protocols used to collect the conditioned medium (CM) then examined for the extracellular expression of the miR. In this case, in which conditions were the cells cultured? How big was the volume of CM used for RNA extraction? Which was the kit used for Total RNA extraction and for miRNA extraction from cells and CM respectively? In the last part of the discussion, the authors mention ( ref 33) a meta-analysis providing evidence that miR-23b-3p downregulation in HCC tissues may be predictor of HCC progression. Now it is known that also the circulating miR-23b-3p in the plasma of HCC patients is downregulated and that sorafenib treatment of HCC patients is associated with longitudinal variation of miR-23b-3p expression detectable in plasma Manganelli M et al Biomedicines. 2021 Jul; 9(7): 813. Published online 2021 Jul 13. doi: 10.3390/biomedicines9070813. These information and reference can be included in the discussion. 

Author Response

Thank you for your valuable comments. We have modified the manuscript as suggested by reviewers. All the queries have been highlighted in the text.

Third Reviewer:

Comment 1.    The authors have chosen the HepG2 cells

HepG2 cell line is liver cancer cell line which is derived from well differentiated HCC patient, it also carry the wild type p53 expression; therefore, it is more closely mimics HCC. HepG2 cells mostly used for research to find level of cytotoxicity and cellular metabolism and drug metabolism in the induced cell line which is required in this study and is fulfilled through HepG2 cells. (Highlighted as suggested; Page 7; Line no. 12)

Comment 2.  Description of miR mimics and antagomiR transfection in HCC cells

Highlighted in Materials and Methods (Page 10; Line no. 12)

Comment 3. The protocols used to collect the conditioned medium (CM) then examined for the extracellular expression of the miR.

Conditioned Medium (CM) for transfection includes:

Lipofectamine RNAiMAX reagent (3 µl) was added in 50 µl of MEM (without antibiotics). Simultaneously, 25 nM of the mimic/antimiR (siRNA) was added in 50 µl of MEM (without antibiotics). Lipofectamine RNAiMAX reagent and siRNA were then added in 1:1 ratio and incubated at RT for 5 mins. For transfection (seed the cells first, followed by transfection), 2 X 105 cells were seeded in 24-well plate to get 60- 80% confluence and then, 50 µl of prepared mixture was added one day prior to experiment. (Highlighted Page 10; Line no. 19)

Comment 4. For extracellular expression, In this case, in which conditions were the cells cultured? How big was the volume of CM used for RNA extraction? Which was the kit used for Total RNA extraction and for miRNA extraction from cells and CM respectively?

Highlighted in Materials and Methods (Page 11; Line no. 1).

Extracellular expression of miRNAs was checked from Media (MEM without antibiotics in HepaG2 parental and resistant cells were grown after the transfection with MiRs and AntiMiRs).

Comment 5. Sorafenib treatment of HCC patients is associated with longitudinal variation of miR-23b-3p expression detectable in plasma.

Modified as suggested (Highlighted Page 19; Line no. 15).

Round 2

Reviewer 1 Report

The authors have now addressed my concerns. 

This manuscript is a resubmission of an earlier submission. The following is a list of the peer review reports and author responses from that submission.

Round 1

Reviewer 1 Report

In this manuscript, the authors undertook an interesting study onSorafenib Resistant Hepatocellular Carcinoma Cells and identified miR23b-3p modulated cytoprotective autophagy and glutamine addiction leads to apoptosis. This is an interesting study to develop new therapeutic strategies for hepatocellular and other cancers which are prone to develop resistance to drugs. However, the present manuscript missing key control experiments.

Major comments:

  1. In general, treatment of drugs such as rapamycin, chloroquine etc needs to be validated by western blots probing for respective target genes such as akt, p-akt, LC3-IIB etc.
  2. Throughout the manuscript western blots of autophagy markers need to be included. Authors made conclusions based only on mRNA studies.
  3. All figures except figure 10 do not have sufficient date to be individual figures. Figures 1,2 can be combined, 3and4 can be combined, 5 and 6 can be combined, 7,8, and 9 can be combined in to one figure.
  4. control experiments for Transfection experiments need to be included.

Minor comments:

  1. Authors need to explain the significance of presence of bright field images of cell morphology in figures.